# The Effect of Ghrelin on the Maturation of Sheep Oocytes and Early Embryonic Development In Vitro

**DOI:** 10.3390/ani12091158

**Published:** 2022-04-29

**Authors:** Daqing Wang, Yanyan Yang, Yongli Song, Shaoyin Fu, Xiaolong He, Biao Wang, Liwei Wang, Xin Chen, Xihe Li, Yongbin Liu, Guifang Cao

**Affiliations:** 1Institute of Animal Husbandry, Inner Mongolia Academy of Agricultural and Animal Husbandry Sciences, Hohhot 010031, China; wangdaqing050789@126.com (D.W.); swallow_0088@163.com (Y.Y.); fushao1234@126.com (S.F.); hexiaolong1983@163.com (X.H.); wangbnd@163.com (B.W.); 18747977760@163.com (L.W.); cx254478@126.com (X.C.); 2School of Veterinary Medicine, Inner Mongolia Agricultural University, Hohhot 010010, China; 3Research Center for Animal Genetic Resources of Mongolia Plateau, School of Life Sciences, Inner Mongolia University, Hohhot 010021, China; songyongli625@163.com; 4The State Key Laboratory of Reproductive Regulation and Breeding of Grassland Livestock, School of Life Sciences, Inner Mongolia University, Hohhot 010021, China

**Keywords:** ghrelin, sheep oocyte, cell cycle, early embryo, RNA-seq

## Abstract

**Simple Summary:**

Different gradients of ghrelin (0, 100, 200, and 300 ng/mL) were added to the IVM system of sheep oocytes to observe their changes, and 200 ng/mL ghrelin was found to be the optimal concentration. The RNA-seq analysis showed that the Cell cycle signaling pathway was enriched. The results suggest that adding ghrelin shortens the duration of IVM of sheep oocytes and hinders early embryonic development.

**Abstract:**

In vitro maturation (IVM) of sheep oocytes and early embryonic development are of great scientific importance for the study of reproductive development in sheep. Ghrelin is an important hormone that regulates the secretion of the growth hormone (GH). In this study, different gradients of ghrelin (0, 100, 200, and 300 ng/mL) were added to the IVM system of sheep oocytes to observe their cell morphology, and Hosesth 33342 staining was used to determine the time taken for oocytes to reach different developmental stages. We found 200 ng/mL ghrelin to be the optimal concentration. The RNA-seq analysis showed that many signaling pathways were significantly altered by ghrelin. Cell cycle, Wnt, and oxidative phosphorylation were activated; the P53 was inhibited. These pathways together regulate the maturation of oocytes and early embryonic development in vitro. The effects of the addition of ghrelin were verified by the expression of GLUT1 in early embryonic development. The results suggest that adding ghrelin shortens the duration of the IVM of sheep oocytes and hinders early embryonic development. This study provides new insights into the effects of exogenous ghrelin on sheep oocyte maturation and early embryonic development in vitro.

## 1. Introduction

Ghrelin, discovered in the 20th century, is a hormone involved in the regulation of growth and has been shown by studies to regulate the secretion of GH [1] in glires and humans. Studies have reported that the receptor for ghrelin (GHSR) is an α-helix 7 transmembrane G protein-coupled receptor that is widely distributed and that ghrelin may have autocrine and compound paracrine effects [2]. Studies over the last decade have demonstrated that GHSR is distributed in the pituitary gland, hypothalamus, stomach, heart, small intestine, lungs, blood vessels, adipose tissue, immune system, multiple central nervous systems, and solid tissues such as human breast tumors [3]. GHSR binding in rat and human brains activates the phospholipase C (PLC) signaling pathway, resulting in an increase in inositol triphosphate (IP3) and the activation of protein kinase C (PKC), which releases calcium stored in cells [4], thereby mediating the growth, development, and metabolism of the organism.

A dose-dependent increase in GH release [5] has been reported in rats by adding different doses of ghrelin. Intravenous administration of ghrelin in rats stimulates gastrin and insulin secretion, and central administration stimulates the release of adrenocorticotropic hormone (ACTH) and prolactin but inhibits thyrotrophin secretion. In human studies, ghrelin not only stimulates the secretion of GH but also promotes the release of ACTH, prolactin, cortisol, and epinephrine, and increases the levels of cyclic adenosine monophosphate (CAMP), but not epinephrine or cyclic guanosine monophosphate (CGMP) [6]. It is evident that ghrelin is not only involved in regulating the secretion of GH but is also closely related to the secretion of other hormones.

Fukuura et al. [7] showed that intra-cerebroventricular injection of ghrelin significantly inhibits the secretion of luteinizing hormone (LH) in ovariectomized rats, so ghrelin may be an important factor in maintaining hormonal homeostasis in vivo.

Recent studies have shown that oocyte IVM is a complex process and that, in addition, these hormones (follicle-stimulating hormone, luteinizing hormone, and estradiol) regulate maturation and have to be added during IVM [8]. How ghrelin affects mammalian oocyte maturation and early embryonic development in vitro needs to be further investigated. Although there are many reports on sheep oocyte culture in vitro, there are still some problems with oocyte IVM compared to normal in vivo oocyte maturation, such as poorer quality, lower fertilization rates, and stalled early embryonic development [9]. Therefore, improving the quality of sheep oocytes cultured in vitro is key to promoting in vitro embryo development.

The differential regulation of early embryonic development by ghrelin has been reported in different animals, confirming that ghrelin is involved in mediating the regulation of mammalian reproductive function [10]. As the mechanism of ghrelin regulation in reproductive function has not been clearly established, the role of ghrelin in mediating reproductive function varies between species. Adding ghrelin at different concentrations to porcine oocytes during IVM culture was effective in increasing the rate of blastocyst development in porcine in vitro fertilization and parthenogenetic embryos [11].

Adding different concentrations of ghrelin to bovine oocyte IVM cultures did not have the same effect, as it effectively accelerated the IVM of oocytes while inhibiting the in vitro development of fertilized embryo blastocysts. In this study, the addition of low concentrations of ghrelin to the IVM of sheep oocytes and an in vitro culture of fertilized ovum increased the in vitro development rate of somatic blastocysts, but high concentrations inhibited the development of ectodermal blastocysts [12]. A previous report showed that the addition of 50 ng/mL ghrelin can increase blastocyst rates. However, adding 250 ng/mL of ghrelin decreased cleavage and blastocyst rates in sheep’s early embryo development. Thus, in this study, the use of 100, 200, and 300 ng/mL ghrelin was explored in sheep’s early embryo development [13].

In this study, the composition of the culture medium for IVM of sheep oocytes was improved, and the mechanism of ghrelin-mediated oocyte IVM was explored by RNA-Seq technology, with the aim of improving the efficiency of livestock breeding and cloning production and providing the most basic and effective guarantee, a new theoretical reference, and research direction for reproductive physiology, in vitro fertilization, embryo development, genetically modified cloning, and livestock breed improvement.

## 2. Materials and Methods

### 2.1. Materials

The ovarian transport method has previously been described [14]. Sheep ovaries were preserved in ovarian transport solution (500 mL normal saline + 150 mL gentamicin/bottle), transported to the lab, and soaked in 75% alcohol for 20 to 30 s. The excess connective tissue around the sheep ovaries was quickly cut off with ophthalmic scissors, followed by three washes with DPBS, and they were then placed in a 60 mm Petri dish containing pre-equilibrated ovotomy solution (90% HM199 + 10% FBS + 2.5 IU Pen Strep + 2.5 IU heparin/mL). Oocytes of 2 to 6 mm in diameter in the ovarian follicles were collected by the cutting method. They were placed in the ovarian washing solution (90% HM199 + 10% FBS + 2.5 IU Pen Strep/mL) and washed and pipetted orally under a stereoscope to select Cumulus Oocyte Complex (COCs) of high quality, namely, normal cells larger than 3 layers of cumulus cells with homogeneous cytoplasm and good morphology were matured and cultured in vitro.

### 2.2. Methods

#### 2.2.1. The Effect of Different Concentration Gradients of Ghrelin on the IVM of Sheep Oocytes

A control group with a basal maturation culture medium (90%M199 + 10%FBS + 15 μg sodium pyruvate + 5 μg cysteamine + 5 μg FSH + 5 μg LH + 5 μg EGF + 5 μg 17-βEstradio + 5 μg gentamicin/mL) was supplemented with 0 ng/mL ghrelin (Peptides, No. PGH-3625-PI); for the experimental group, 100, 200, and 300 ng/mL ghrelin were added to the basal maturation culture medium. The oocytes in the two groups were matured for 22 h at 38.5 °C in a 5% CO_2_ saturated humidity incubator and collected from the in vitro cultures.

The oocytes of the control and experimental groups were counted to determine the number of COCs after pre-treatment with 200 μL hyaluronidase for 3 min at 38.5 °C and 5% CO_2_ in a saturated humidity incubator and then gently blown off the cumulus cell. The number of oocytes with an extruded polar body (number of oocytes at the MII stage) was counted, and the maturation rate of the control and experimental groups were calculated. The sample size was *n* = 350, and procedures were performed three times.

#### 2.2.2. Fixation and Live Cell Staining

The cells used for staining were all living cells. The staining did not involve cell fixation, but the cells were placed on the glass slide, then the four corners were supported with paraffin wax, and the glass slide was gently covered, then sealed with a neutral resin glue, and photographed for observation.

#### 2.2.3. Investigating the Changes in the Nuclear Morphology of Oocytes over Time under an Exogenous Addition of Ghrelin

By adding 200 ng/mL ghrelin to sheep oocytes during maturation, 5 mg/mL Hoechst was used as the cultural medium and was stained for 20 min. Finally, stained cells were removed and cleaned 3 times in DPBS. The slide was placed in the center and sealed with neutral gum. Changes in the nuclear morphology of the oocytes during in vitro nuclear maturation were observed, and the different stages of oocyte nuclear maturation (GV, GVBD, MI, MII) were determined by changes in the nuclear morphology.

#### 2.2.4. Real-Time PCR

The RXeasy Mini Kit (Qiagen, 74104) was used in this study. Expression changes of DNA45 (GADD45), Cyclin B1 (CCNB1), and GLUT1 at different stages of oocyte nuclear maturation were measured using qPCR and qPCR primer, as shown in Table 1.

Individual cellular RNA was extracted by the exogenous addition of 200 ng/mL ghrelin and without the addition of ghrelin at different stages of oocyte nuclear maturation.

#### 2.2.5. The Effect of Ghrelin on the Fertilization and Early Embryonic Development of Sheep Oocytes In Vitro

Sheep oocytes were cultured in vitro by adding 200 ng/mL ghrelin during maturation; frozen sperm with a sperm viability higher than 60% was selected (Inner Mongolia SK·Xing Animal Breeding and Breeding Biotechnology Research Institute Co., Hohhot, China). The sperm were washed twice (309× *g*, 3 min) by centrifugation in fertilization solution A (IVF-A, 0.78 mL embryo water (Sigma, W1503-500 mL) + 100 μL stockA + 100 μL stockB + 10 μL stockC + 10 μL stockD + 0.005 g BSA + 4 IU heparin/mL), activating the sperm, for which the sperm may be able to prepare. The sperm were resuspended to 10 million/mL with IVF-B (0.78 mL embryo water + 100 μL stockA + 100 μL stockB + 10 μL stockC + 10 μL stockD + 20% FBS + 4 IU heparin/mL) and diluted to 10 million/mL in each of the equilibrated droplets of the capacitation solution, and then 50 μL of diluted sperm was added to the equilibrated droplets of the capacitation solution and incubated in a CO_2_ incubator for 30 min to capacitate the sperm.

Mature oocytes were washed in groups of 15 to 20 with IVF-B and placed in diluted sperm droplets that had been capacitated. After 22 h, the fertilized oocytes were removed from the incubator, placed in the cumulus cell solution, washed 3 times, and then transferred by pipetting into a pre-warmed Sof (Medium Semen In Vitro Fertilization) solution (0.718 mL embryo water +100 μLstockA + 100 μL stockB + 10 μL stockC + 10 μL stockD + 10 μL stockX + 10 μL stockY + 10 μL stockY + 10 μL stockZ + 10 μL MEM non-essential amino acid 100X+ 20 μL MEM essential amino acid 50X0.008 g BSA + 4 IU Pen Strep/mL) in an incubator with a CO_2_ level of 5%, humidity of 100%, and at a temperature of 38.5 °C for 7 days, with 4% FBS added on day 5 to ensure adequate nutrition for embryo development. After fertilization, we checked the embryo number of the 2-cell, 4-cell, 8–16-cell, morula, and blastocyst, and then the changes in ghrelin-mediated oocyte cleavage in early embryos were observed over time. The liquid ingredient formulation standard is shown in Table 2.

#### 2.2.6. Statistical Analysis

Unless otherwise noted, all the analyses were performed in at least triplicate. A Student’s *t*-test was performed using Chi-share-test. *p* < 0.05, *p* < 0.01, and *p* < 0.001 were considered statistically significant. All the data are reported as mean ± SD. The means and standard deviations from at least three independent experiments are represented in all the graphs.

## 3. Results

### 3.1. Effect of Ghrelin on the Maturation Rate of Sheep Oocytes in the MII Stage

A previous report showed that the addition of 50 ng/mL ghrelin can increase blastocyst rates. However, adding 250 ng/mL of ghrelin decreased cleavage and blastocyst rates in sheep’s early embryo development. Thus, in this study, the use of 100, 200, and 300 ng/mL ghrelin was explored. The maturation rates of oocytes were calculated by adding different concentrations of exogenous ghrelin: 69.4% (*p* < 0.05) for 100 ng/mL ghrelin, 85.6% (*p* < 0.01) for 200 ng/mL ghrelin, and 75.9% (*p* < 0.05) for 300 ng/mL ghrelin. The addition of 100, 200, and 300 ng/mL ghrelin had a beneficial effect on oocyte maturation during IVM of sheep oocytes, with 200 ng/mL ghrelin having a highly significant effect (*p* < 0.01) and 300 ng/mL ghrelin and 100 ng/mL ghrelin having a significant effect (*p* < 0.05), as shown in Table 3.

### 3.2. Exploring the Changes in the Nuclear Morphology of Sheep Oocytes with Time under the Effect of Ghrelin

Schematic diagram of oocyte nuclear maturation and changes in the morphology of oocyte nucleus maturation by Hoechst 33342 staining during in vitro maturation of oocytes in control and experimental groups. Ghrelin did not alter the morphological changes of the nucleus maturation during in vitro maturation of oocytes, as shown in Figure 1.

Control: oocytes matured in vitro at 0 h: 97% reaching the GV stage (*p* < 0.01), at 8 h: 52.46% reaching the GVBD stage (*p* < 0.01), at 16 h: 54.54% reaching the MI stage (*p* < 0.01), and at 22 h: 56.42% reaching the MII stage (*p* < 0.01). Treated: oocytes matured in vitro at 0 h: 97% reaching GV stage (*p* < 0.01), at 6 h: 62.77% reaching GVBD stage (*p* < 0.01), at 14 h: 66.84% reaching MI stage (*p* < 0.01), and at 16 h: 64.03% reaching MII stage (*p* < 0.01). Ghrelin accelerates the maturation of oocytes in vitro, as shown in Table 4 and Table 5.

Expression of cyclin B1 and GADD45 at different stages of oocyte maturation. The expression of GADD45 and cyclin B1 in the mature GVBD stage of oocytes in the experimental group and control group was significantly different (*p* < 0.01). After ghrelin addition, GADD45 and cyclin B1 are expressed differentially, mainly in vitro maturation culture for 6–8 h, as shown in Figure 2.

### 3.3. RNA-Seq Analysis of the Molecular Mechanisms of Oocyte Maturation in Sheep Following the Addition of Ghrelin

Data analysis: RNA-seq single-cell library preparation and sequencing. The raw sequences were obtained by sequencing and contained joints and low-quality reads. The raw reads were filtered to ensure quality, and valid samples of clean reads more than 5 G were obtained, with Q20 > 99% and Q30 > 83.66%. The proportion of valid reads exceeded 99.75%, and the GC content was more than 41.5%. The sequencing data quality was good and could be used for subsequent data analysis (Biotechnology Co., Ltd., Hangzhou, China).

To understand the mechanisms of molecular regulation in ghrelin-mediated sheep oocytes during the cell division cycle, gene expression was determined by RNA-seq. The volcano plots showed significant differences in the cell division cycle of the exogenously added ghrelin group and the control group, with 3587 differentially expressed genes (DEGs) (*p* < 0.05, fold change >10) in the control group at 8 h (OMM08h) and the exogenously added ghrelin group matured in culture in vitro at 6 h (MGM06h), of which 1535 genes were upregulated and 2052 genes were downregulated. There were 1505 DEGs between the control group at 16 h (OMM16h) and the exogenously added ghrelin IVM culture at 14 h (MGM14h) (*p* < 0.05, fold change >10), of which 727 genes were upregulated and 778 genes were downregulated. There were also 1667 DEGs in the control group at 22 h (OMM22H) and in the exogenously added ghrelin IVM culture at 16 h (MGM16h) (*p* < 0.05, fold change >10), of which 697 genes were upregulated and 970 genes were downregulated. There were 1667 DEGs (*p* < 0.05, fold change >10) in the control group at 22 h (OMM22H) and in the exogenously added ghrelin group matured in vitro (MGM16h), of which 697 genes were upregulated and 970 genes were downregulated, as shown in Figure 3.

IVM sheep oocytes exogenously supplemented with ghrelin were significantly enriched for genes of the cell cycle, oxidative phosphorylation, and Wnt; signaling pathways were activated at 6 h, 8 h, and 16 h of IVM. Genes P53 signaling pathways were suppressed in the gene set enrichment analysis (GSEA) as compared to the control group without the addition of the exogenous agent, as shown in Figure 4.

Next, ribosome synthesis (DNAJC9, MRPS16), the generation of oxidative phosphorylation (ATP5A1, ATP5D), the cell cycle (SMAD4, CDC25, CDK6, WEE2, GDAA45, Bub2), RNA transport (CYFIP2, DDX20), the metabolism of inositol phosphate (MINPP1, PIK3C2A), meiosis of oocytes (AURKA, CALM1, CCNE2, CDC25A, CDC20, PLK1), the PI3K-AKT signaling pathway (BAD, CDKN1A), and other differentially significant genes were analyzed by thermography. The analysis revealed that the addition of exogenous ghrelin had the strongest relationship with intercellular oxidative phosphorylation, DNA replication, cell cycle regulation, and oocyte meiosis, with a high number of differentially expressed genes, as shown in Figure 5.

### 3.4. The Effect of Ghrelin on In Vitro Fertilization and Early Embryonic Development of Sheep Oocytes

Early embryo development was observed under a stereomicroscope. The early cleavage of the embryos of the control group (without exogenous supplementation of ghrelin, OMM) and the experimental group (with exogenous supplementation of ghrelin, MGM) were observed at 22 h, 48 h, 72 h, 96 h, 120 h, and 5 d of in vitro incubation, and the early cleavage rates were determined as follows. Experimental group (treated): 70% of sheep oocytes fertilized in vitro cleaved to 2 cells at 22 h (*p* < 0.05); 65% cleaved to 4 cells at 48 h (*p* < 0.05); 56% cleaved to 8 cells at 72 h (*p* < 0.05), 35% cleaved to 16 cells at 96 h (*p* < 0.05); 17% cleaved to morulas at 120 h (*p* < 0.05); and 12% cleaved to blastocysts at 5 d (*p* < 0.05). The cleavage rate in the experimental group was 70% (*p* < 0.01). Control group (control): 80% of sheep oocytes fertilized in vitro cleaved to 2 cells at 22 h (*p* < 0.05); 78% cleaved to 4 cells at 48 h (*p* < 0.05); 65% cleaved to 8 cells at 72 h (*p* < 0.01); 26% cleaved to 16 cells at 96 h (*p* < 0.05); 20% cleaved to morulas at 120 h (*p* < 0.05); and 17% cleaved to blastocysts at 5 d (*p* < 0.05), with a cleavage rate of 80% in the experimental group (*p* < 0.01). Ghrelin did not mediate changes in early embryonic development after the fertilization of sheep oocytes, as shown in Figure 6 and Table 6.

GLUT1 was expressed to varying degrees during early embryonic development, with highly significant differences (*p* < 0.01) between the treated and control groups when the cells reached 8-cell, 16-cell, and the morula stages of cleavage during development, as shown in Figure 7.

## 4. Discussion

The IVM of sheep oocytes, as a fundamental part of bioengineering research such as in vitro fertilization, embryo development, and genetically modified cloning, is mediated not only by the intrinsic regulation of its own secretory factors but also by exogenous factors. It has been shown that ghrelin is mediated by binding to G protein-coupled receptors [15]. Studies [16] on the involvement of ghrelin in regulating oocyte maturation and embryo development have shown that adding ghrelin at different gradients during oocyte embryo culture can affect the maturation state of oocytes and even blastocyst development, thus influencing the regulation of the cell cycle.

There are few studies on whether the regulation of sheep’s reproductive development by ghrelin begins at the maturation stage [17], and its optimal dose has also not been reported. In this study, ghrelin was added to IVM sheep oocyte cultures at different doses for the first time. In this study, differences in oocyte COCs, MII oocyte numbers, and oocyte maturation rates after 22 h of IVM with the addition of different concentrations of ghrelin to IVM sheep oocyte cultures were observed.

The results showed that different concentrations of ghrelin mediated the increase in the sheep oocyte IVM rate, which can be seen as: 200 ng/mL (*p* < 0.01) > 300 ng/mL (*p* < 0.05) > 100 ng/mL (*p* < 0.05). The results indicated that different concentrations of exogenous ghrelin mediated the maturation process of sheep oocytes in vitro, which is consistent with the findings of Mstias et al. [18].

The RNA-seq analysis revealed that adding the optimal dose of ghrelin had a significant effect on the IVM of oocytes, and that cell cycle regulation was intricately related to oxidative phosphorylation and the P53 and Wnt signaling pathways, with varying degrees of transmission mediated and reciprocal effects between the signaling pathways [19]. It is hypothesized in this study that ghrelin’s effect on cell proliferation is mainly mediated through the cell cycle, Wnt, and P53 signaling pathways.

In summary, ghrelin positively and effectively regulates sheep oocyte IVM culture, in which Wnt, P53, and cell cycle signaling pathways are essential as bridges for transmission cross-talk, especially at MOM16h as compared to MGM14h, where the differential expression of the P53 signaling pathway is most significant, and they do not alter cellular progression while prematurely blocking cell cycle progression. In this study, we demonstrated for the first time that ghrelin mediates the cell cycle regulation of sheep oocyte IVM through multiple signaling pathways at different stages of IVM. Ghrelin accelerated IVM significantly.

Recent studies have shown that ghrelin may be involved in the regulation of physiological functions in a variety of tissues [6] and that its binding to its receptor has many functions. It can effectively regulate the synthesis and release of GH, regulate energy metabolism, and has a role in early embryonic development and the secretion of reproductive hormones. In particular, the ghrelin expression of cervix uteri, oviduct, and early oocyte in mammals and the exploration of their associated mechanisms have led to a steady stream of highly credible evidence that ghrelin actively mediates mammalian reproduction-related activities. Ghrelin is currently being increasingly studied in the reproductive system. In this study, mature sheep oocytes were obtained by adding 200 ng/mL ghrelin to the IVM culture of sheep oocytes, and the mature oocytes were cultured for in vitro fertilization. Early embryo development after fertilization was observed and measured after 22 h. The results showed that adding 200 ng/mL ghrelin during the in vitro fertilization culture inhibited the development of blastocysts in vitro. Recently, it has been reported that ghrelin plays an important role in early embryonic development [20]. Additionally, Zhang et al. [21] inhibited oocytosis of embryos at the 2-cell stage towards hatching blastocysts and suppressed changes in their developmental rate. However, the inhibition was significantly relieved by the re-addition of an antagonist of the ghrelin receptor, GHSR-1α. This study explored the possibility that the addition of 200 ng/mL ghrelin to mature sheep oocytes in vitro may have crossed a low concentration threshold during later in vitro fertilization and did not effectively improve the cleavage rate of early embryos after in vitro fertilization, a result consistent with Torres’ study, in which the addition of 200 ng/mL ghrelin tended to positively affect oocyte maturation.

GLUT1, a glucose transporter protein of the cotransporter superfamily, has been found to be involved in the metabolism of oocytes and early embryos [22]; ghrelin is involved in the energy metabolism of the early embryos after oocyte fertilization in vitro by mediating the differential expression of the GLUT1 gene at different developmental stages of the early embryo, which, in turn, mediates the quality and further developmental capacity of the early embryo through the energy metabolism pathway.

It is hypothesized that ghrelin mediates the downregulation of GLUT1 gene expression at different stages of early embryonic development, resulting in insufficient energy supply at later stages of embryonic development, further reducing the quality of early embryos and their ability to develop further.

In this study, ghrelin-mediated IVM of sheep oocytes was used to obtain mature oocytes for in vitro fertilization experiments and to explore early embryonic development.

## 5. Conclusions

The addition of 200 ng/mL ghrelin can reduce the duration of oocyte maturation. However, it did not reduce the duration of embryo development. Cell cycle, oxidative phosphorylation, and Wnt signaling pathways were activated. The P53 signaling pathway was inhibited. This study provides new insights into the effects of exogenous ghrelin on sheep oocyte maturation and early embryonic development in vitro and provides new ideas regarding the cell cycle pathway in the regulation of IVM of oocytes.

## Figures and Tables

**Figure 1 animals-12-01158-f001:**
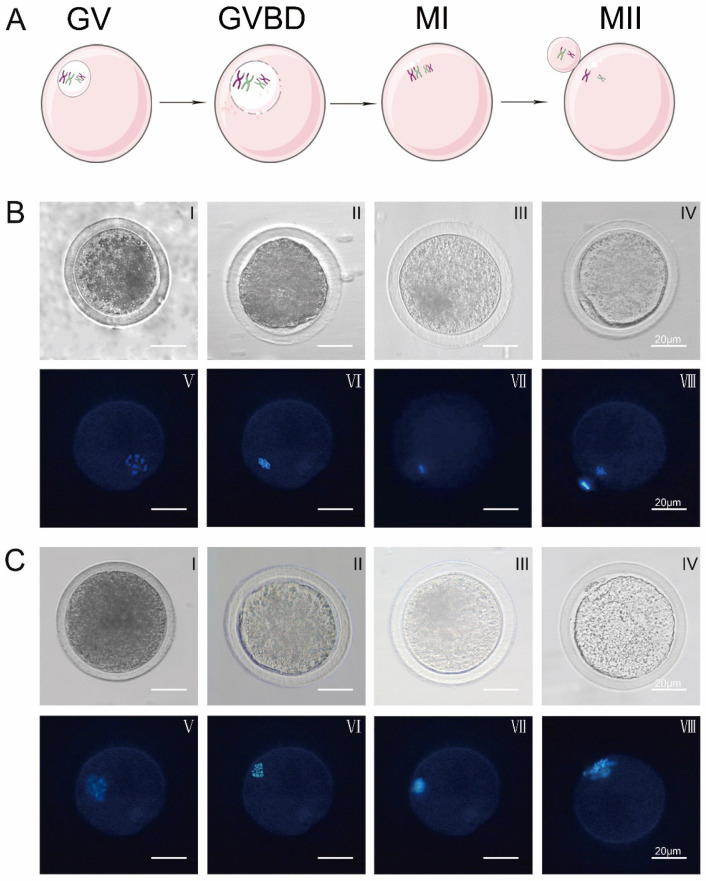
Changes in oocyte nuclear morphology over time in response to ghrelin. (**A**) Diagram of different stages of IVM oocytes (GV, GVBD, MI, MII). (**B**) Control group: (0 ng/mL) oocyte nuclear maturation cell morphology (I–IV) and Hoechst 33342 stained cell morphology (V–VIII) (scale bar: 20 μm) (**C**) Experimental group: (200 ng/mL) oocyte nuclear maturation cell morphology (I–IV) and Hoechst 33342 stained cell morphology (V–VIII) (scale bar: 20 μm).

**Figure 2 animals-12-01158-f002:**
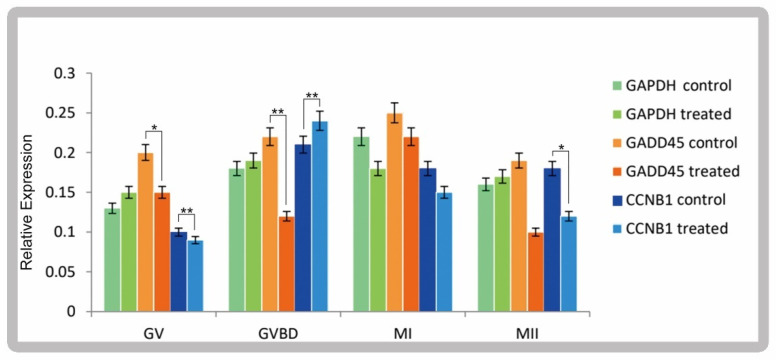
Relative expression of GADD45 and CCNB1 at different stages of IVM oocyte. Error bars indicate three independent biological replicates (mean ± standard deviation). * *p* < 0.05, ** *p* < 0.01.

**Figure 3 animals-12-01158-f003:**
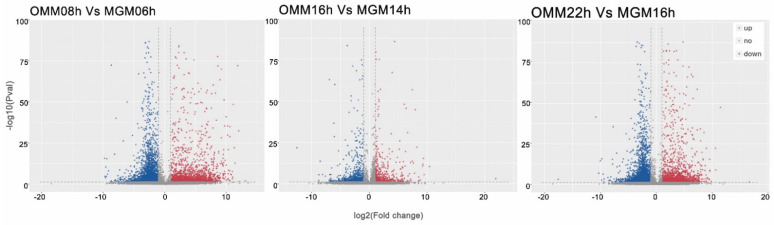
Volcano plot of differentially expressed genes at different time points. Red represents genes that are significantly different and upregulated; blue represents genes that are significantly different and downregulated; gray represents genes that are not significantly different.

**Figure 4 animals-12-01158-f004:**
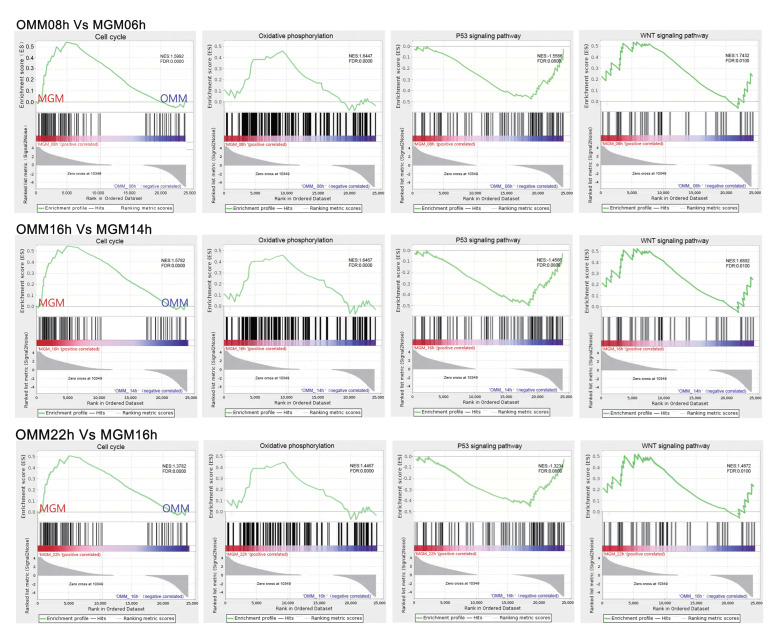
Gene set enrichment analysis (GSEA) of different time points. The green line shows the enrichment profile. NES: normalized enrichment score, FDR: false discovery rate.

**Figure 5 animals-12-01158-f005:**
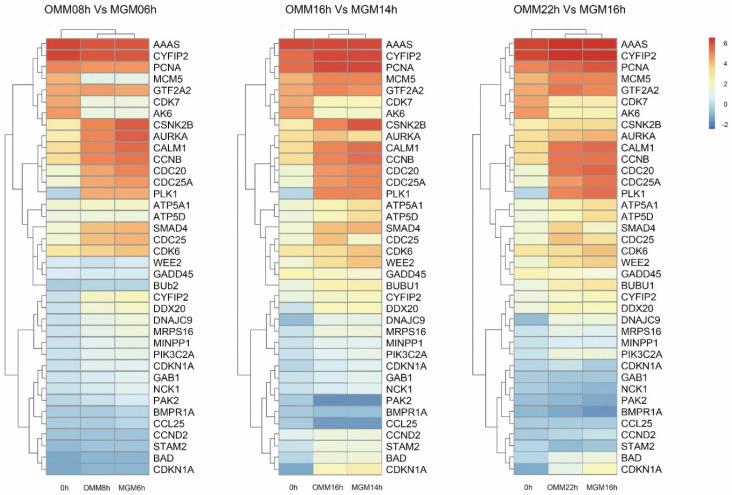
Heatmap showing the expression comparison of markers at different time points. Ribosome synthesis, generation of oxidative phosphorylation, cell cycle, RNA transport, metabolism of inositol phosphate, meiosis of oocytes, PI3K-AKT signaling pathway, and other differentially significant genes.

**Figure 6 animals-12-01158-f006:**
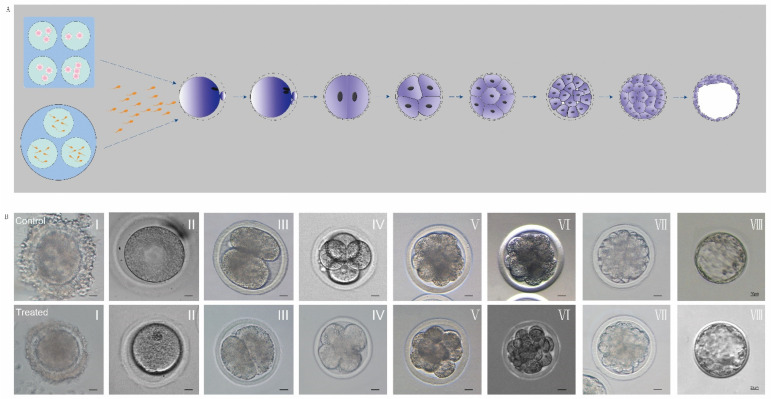
Effect of ghrelin addition on early embryonic development in sheep. (**A**) Diagram of fertilization and early embryonic development in vitro. (**B**) Cell morphology during fertilization and early embryonic development in vitro. I. Fertilization, II. Fertilization finish, III. 2 cell, IV. 4 cell, V. 8 cell, VI. 16 cell, VII. Morula, VIII. Blastocyst. Scale bar: 20 μm.

**Figure 7 animals-12-01158-f007:**
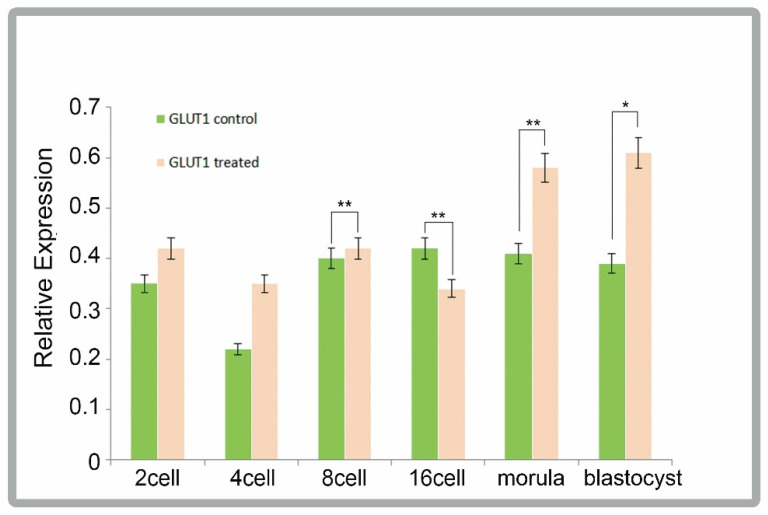
qPCR analysis of GLUT1 in different development stages. Error bars indicate three independent biological replicates (mean ± standard deviation). * *p* < 0.05, ** *p* < 0.01.

**Table 1 animals-12-01158-t001:** Quantitative PCR primer sequences.

Target Gene	Primer Sequences
GAPDH	F:5′-CCTGAGACAAGATGGTGAAGG-3′;
	R:5′-ATGGGTGGAATCATCATGGAAC-3′
GADD45	F:5′-TGCTACTGGAGAACGACGC-3′;
	R:5′-GGATCCTTCCATTGTGATGAA-3′
CCBN1	F:5′-CACAGGATACACAGAGAATG-3′
	R:5′-CTTGATGGCGATGAATTTAG-3′
GLUT1	F:5′-CATGTGCTTCCAGTATGTGGAG-3′
	R:5′-CTCAGGTGTCTTGTCACTTTGG-3′

**Table 2 animals-12-01158-t002:** Base Liquid ingredient formulation standard.

Title	Ingredients (Article No., Manufacturer)	Mass/Volume	Concentration
StockA	NaCl (S5886, sigma) KCl (p5405, sigma) KH_2_PO_4_ (p5655, sigma) MgCl_2_·6H_2_O (m2393, sigma) Na-lactate (l7900, sigma) Glucose (g7021, sigma)	1.166 g 0.1068 g 0.0324 g 0.02 g 0.1232 mL 0.054 g	997.6 mM 71.63 mM 11.90 mM 4.919 mM 0.616 (*v*/*v*)% 14.99 mM
	Ultrapure water	20 mL	
StockB	NaHCO_3_ (S5761, sigma)	0.42 g	250.0 mM
	Ultrapure water	20 mL	
StockC	Na Pyruvate (p4562, sigma)	0.018 g	32.72 mM
	Ultrapure water	5 mL	
StockD	CaCl_2_·2H_2_O (c7902, sigma)	0.1008 g	171.4 mM
	Ultrapure water	5 mL	
StockX	L-Glutamine (g3126, sigma)	0.073 g	99.90 mM
	Ultrapure water	5 mL	
StockY	Sodium citrate (s1804, sigma)	0.05 g	34.00 mM
	Ultrapure water	5 mL	
StockZ	Myo-Inositol (I7508, sigma)	0.2495 g	277.0 mM
	Ultrapure water	5 mL	

**Table 3 animals-12-01158-t003:** Effect of different concentrations of ghrelin on oocyte information rate.

Concentration of Ghrelin (0 ng/mL)	Total Number of COCs (Number of Repetitions)	Number of Oocytes in the MII Phase	Maturation Rate
0	350 (3)	223	63.7%
100	350 (3)	243	69.4% ^b^
200	350 (3)	300	85.6% ^a^
300	350 (3)	266	75.9% ^b^

Note: ^a^ indicate significance (*p* < 0.01), ^b^ indicate significance (*p* < 0.05). Three independent biological replicate statistics (n = 3, mean ± standard deviation).

**Table 4 animals-12-01158-t004:** Number of cells at different stages of IVM oocyte in control groups.

Incubation Time	Number of Oocytes (Number of Repetitions)	GV	GVBD	MI	MII
0 h	400 (n = 3)	(97.00 ± 2.66) ^a^	(3.00 ± 0.66) ^b^	0	0
8 h	400 (n = 3)	(14.00 ± 3.17) ^b^	(52.46 ± 1.66) ^a^	(18.46 ± 0.75) ^b^	(15.08 ± 2.66) ^b^
16 h	400 (n = 3)	(4.46 ± 2.66) ^b^	(52.46 ± 1.67) ^b^	(54.54 ± 2.66) ^a^	(26.16 ± 2.06) ^b^
22 h	400 (n = 3)	0	(52.46 ± 1.68) ^b^	(37.33 ± 5.66) ^b^	(56.42 ± 2.01) ^a^

Note: ^a^ indicate significance (*p* < 0.01), ^b^ indicate significance (*p* < 0.05). Three independent biological replicate statistics (n = 3, mean ± standard deviation).

**Table 5 animals-12-01158-t005:** Number of cells at different stages of IVM oocyte in experimental groups.

Incubation Time	Number of Oocytes (Number of Repetitions)	GV	GVBD	MI	MII
0 h	400 (n = 3)	(97.00 ± 2.01) ^a^	(7.46 ± 1.06) ^b^	0	0
6 h	400 (n = 3)	(13.33 ± 3.17) ^b^	(62.77 ± 1.66) ^a^	(11.62 ± 0.75) ^b^	(12.28 ± 2.66) ^b^
14 h	400 (n = 3)	0	(9.29 ± 2.01) ^b^	(66.84 ± 1.37) ^a^	(23.87 ± 2.00) ^b^
16 h	400 (n = 3)	0	0	(35.97 ± 4.06) ^b^	(64.03 ± 2.91) ^a^

Note: ^a^ indicate significance (*p* < 0.01), ^b^ indicate significance (*p* < 0.05). Three independent biological replicate statistics (n = 3, mean ± standard deviation).

**Table 6 animals-12-01158-t006:** Number of early cleavage of embryos.

Sample	Number of Fertilized Oocytes (Number of Replicates)	2 Cell	4 Cell	8 Cell	16 Cell	Morula	Blastocyst	Cleavage Rate
		Number (% ± SEM)						
Control	343 (n = 3)	275 (80 ± 4.50)	267 (78 ± 4.00)	233 (65 ± 2.00) ^b^	89 (26 ± 1.52) ^b^	69 (20 ± 1.73) ^b^	58 (17 ± 1.70) ^a^	80%
Treated	352 (n = 3)	264 (70 ± 5.29)	229 (65 ± 5.29)	197 (56 ± 2.51) ^b^	123 (35 ± 1.52) ^b^	60 (17 ± 1.73) ^b^	42 (12 ± 2.00) ^a^	70%

Note: ^a^ indicate significance (*p* < 0.01), ^b^ indicate significance (*p* < 0.05). Three independent biological replicate statistics (n = 3, mean ± standard deviation).

## Data Availability

The data used to support the findings of this study are available from the corresponding author upon request.

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
