# Peer review of "The Effect of Ghrelin on the Maturation of Sheep Oocytes and Early Embryonic Development In Vitro"

_animals, 2022, doi:10.3390/ani12091158_

Round 1

Reviewer 1 Report

I congratulate authors because of their work, but some issues are missing and should be improved.

General comments:

  • Format and english revisions are highly recommended
  • Introduction should be improved to attract reader interest 
  • M&M minimal improves and references
  • Results: improve clarity and include tables. I found no sample size (N), nor numer of voaries, oocytes, etc. Please, review.
  • Discussion is too long 

Some other comments:

Simple summary is the same than abstract. Please rewritte. 

Abstract must be more clear, with less acronysms. 

The first time you use an acronysm, please, say the meaning. Do not start sentences with acronysms.

Introduction: I missed more references thourghout the text when explaining contents. Is there no evidence of ghrlein use in sheep in IVM? Could you explain maybe why? Could you provide some results from other species that encourage you to do this experiment?

Ethical statement maybe at the end of the article, not in a separate section in M&M.

Material 2.2 please provide references of this technique as it is validated in other studies. Please specify number of ovaries, of sheeps, if the collection was at a time or at different times, etc. More information about this part is necessary.

References are missing throughout the whole M&M you described. I guess these protocols have been previously validated.

ln 175-179 rewritte to enhance clarity

ln 190-195. Difficult to follow. I understand that figures D and E are very beautiful and visual, but I would rather tables to follow properly the results and data obtained.

ln 257-270. Maybe a table would be easier to follow these results.

Discussion is too long. Please, summarize it and shorten it.

ln 309-314 are not necessary.

ln 319. here you mention the study of Mstias and colleagues, who did it in bovine, maybe including some information in the introduction is welcome.

It would be welcome to speak about the impact of using this hormone in these procedures highlighting the impact (or not) that coudl have in embryo cryopreservation, embryo implantation and pregnancy loss. Is there any evidence of this?

Author Response

Thank you very much,reply to attachment

Reviewer 2 Report

Review animals-1673256

In this study, the influence of Ghrelin on in vitro maturation of sheep oocytes and early embryo development is investigated. Gene expression analysis was performed to better understand the molecular regulation of Ghrelin supported maturation.

This manuscript first of all needs a thorough improvement of the language. The scientific value of the study is overshadowed by the many spelling and punctuation errors, making many sentences very difficult to understand. Many phrases are imprecise or do not make sense, and the common thread is not always apparent.

Specific comments:

Introduction

The sheep is in an important livestock and model animal and the improvement of assisted reproduction is therefore of great relevance. Unfortunately, this study lacks a brief summary of the current state of the art of assisted reproduction in sheep.

In general, I would focus more on in vitro maturation in sheep in the introduction to give the reader an idea what are common in vitro maturation and fertilization rates in sheep, rather than explaining the functions of Ghrelin that are irrelevant to reproduction. Also, the reader should be better prepared on why you performed gene expression analysis and what your specific focus was here. 

Just two examples of language troubles:

Line 46: Human breast tumors are not peripheral tissue!

Line 63-65: IVM is NOT the process of removing immature COCs from the ovary....

Material and Methods

Please explain how you performed Hoechst staining. Did you perform fixation? It sounds more like you tried live cell staining. Did you stain the oocytes several times at different time points?

Why did you choose different time points for control and treatment groups?

How did you decide how much Ghrelin needs to be used? The difference between 100, 200 and 300ng is not very high.

Line 143/154: What is Embryo water? Please define also your stock solutions (in a table maybe)!

Statistics: I think a Chi-Share-test would be correct to compare maturation rates between treatment groups and control.

As far as I could see you have not done this comparison.

Results

Fig. 1B: To me it looks like there cannot be a significant difference between 0; 100 and 300ng treatment. Please check again. Diagram explanation is too small to read.

I don´t understand what Fig. 1A should show? I think it is not necessary.

Fig. 2: Figures are too small to really read it. Scale bar is probably wrong. Are you sure that the diameter of sheep oocytes is at least 200µm? Human oocytes, for example, are around 100µm.

I don`t understand why you choose these time points to assess diffenent maturation stages.

Fig  3 and 4: too small.

Fig 4: what is the message?

Line 255 ff: Could you provide a table of embryo development? It would be much easier to understand it.

Fig. 6: Scale bar is probably incorrect

Discussion

Some parts of the discussion are very detailed but difficult to understand. The authors should define their results more clearly and then focus their discussion on the essential findings. A diagram of the regulatory mechanisms and pathways would be helpful for understanding.

Appendix

Data Availability Statement: incorrect (hopefully)

Conflict of interest is actually author contribution, except the last sentence.

The value of the study is severely diminished by the language problems. The manuscript cannot be published in this form. It needs extensive language improvement. This would also help the authors to elaborate their results more precisely and to make the discussion more concise but understandable.

Author Response

Thank you very much,reply is attachment

Round 2

Reviewer 1 Report

I would like to congratulate authors. They did a hard-work on the manuscript and the result is very positive. 

When including the sample size, I think is a bit confusing saying the sample size and then n=3 (repetitions). Maybe it is better to say just the sample size (n) and in M&M desribe which procedures have been performed three times, or something like that. Because your sample size is higher than 3, it is just the times you replicate.

I suggest to include under each tabe or figure a legend where every acronym is specified (for example, I found no COC description in the manuscript). As well, I suggest to increase size of some figures which can not be visualized properly.

In the section "Conflicts of interest", I think it should be written something like: authors declare that there are no confilcts of interest" or something like that.

Author Response

We thank the editor and reviewer1 for their enthusiasm and  many insightful comments on our manuscript (1673256). We have comprehensively addressed all of the comments from Reviewer1. We sincerely hope that the revised and improved manuscript is now suitable for publication in Animals Below please find our detailed point-by-point response to the Reviewers.

Reviewer 2 Report

Dear authors,

Please find my new comments below in brown color. I think the manuscript has improved dramatically. However, there are still many issues that need to be addressed. In general all tables look better and more professional in the word document than in the pdf document.

Abstracts: I marked some write errors.

Major comments:

Introduction

1.The sheep is in an important livestock and model animal and the improvement of assisted reproduction is therefore of great relevance. Unfortunately, this study lacks a brief summary of the current state of the art of assisted reproduction in sheep.

Response: We thank the reviewer for his/her suggestion. We have added ghrelin role in other species in introduction.Due to assisted reproduction in sheep research were not performed.However,this is a very good issue. We will perform effect of ghrelin on assisted reproduction in sheep in the future.

2.In general, I would focus more on in vitro maturation in sheep in the introduction to give the reader an idea what are common in vitro maturation and fertilization rates in sheep, rather than explaining the functions of Ghrelin that are irrelevant to reproduction. Also, the reader should be better prepared on why you performed gene expression analysis and what your specific focus was here. 

Response: We thank the reviewer for his/her suggestion. We have rewritten introduction,and added ghrelin role in other species.We wondered that which signaling pathway and gene expression were changed after adding ghrelin.

3.Line 46: Human breast tumors are not peripheral tissue

Response: We thank the reviewer for his/her suggestion. We have change this wrong in revised version.

4.Line 63-65: IVM is NOT the process of removing immature COCs from the ovary

Response: We thank the reviewer for his/her suggestion. We have deleted this paragraph with help of many experts.

Line 63-64: I think you need a clear definition of what endogenous and exogenous factors are. Endogenous hormones have to be replaced during IVM, by adding them to the maturation medium. So are they then exogenous? Maybe avoid these terms and say that theses hormones regulate maturation and have to be added during IVM.

Line 77: the correct term in mammals is blastocyst and not blastula (thoug “morula” is correct. Please replace in the whole manuscript.

Material and Methods

  1. Please explain how you performed Hoechst staining. Did you perform fixation? It sounds more like you tried live cell staining. Did you stain the oocytes several times at different time points?

Response: We thank the reviewer for his/her suggestion.

Line 121: the usual term is “oocytes with extruded polar body…”

Line 122: You are not counting rates, you are counting numbers und then calculating maturation or fertilization rates

5mg/mL Hoechst was oocytes cultural medium oocytes and stained for 20min. Finally, stained cells were removed and cleaned 3 times in DPBS. The slide was placed in the center of the slide and was sealed with neutral gum.

You need to add this information to the Material and Methods!

Fixation and live cell staining

The cells used for staining were all living cells. The staining did not involve cell fixation, but the cells were placed on the glass slide, then the four corners were supported with paraffin wax, and the glass slide was gently covered, then sealed with neutral resin glue, and photographed for observation

You need to add this information to the Material and Methods!

Line 127/128: please edit the information is doubled.

Line 130: PCR topic is now under the headline of oocyte morphology analysis. That does not make sense.

Line 143: please add g number! Rpm vary depending on the centrifuge you are using

Line 145: after the heparin concentration I would already add the marked sentence of line 163.

Line 155: SOF, abbreviation for?

Line 160/161: After 48 h you are not counting all of these stages! Also, I think elsewhere you said you checked after 24h, 48h and so one…

Staining time point selection:

The staining experiment aims to observe the morphological changes of the nuclei during the maturation of oocytes, stain the experimental group and the control group, the experimental group adds 200 ng/ml Ghrelin group to the in vitro maturation culture process, and the control group adds 0ng/ml Ghrelin group to the in vitro maturation culture process, and after 0h, 6h, 8h, 14h, 16h, 22h in vitro maturation culture, it is taken out of the Petri dish for staining and photographic observation.120 oocytes were retrieved for each experiment, and the experiment was repeated 3 times per time period, after which the results were recorded.

6.Why did you choose different time points for control and treatment groups?

Response: We thank the reviewer for his/her suggestion. Oocyte in vitro maturation can be visually determined by nuclear staining, and the control group and experimental group oocytes reach the GV, GVBD, MI and MII stages of oocyte maturation at 0h, 8h, 16h, 22h and 0h, 6h, 14h and 16h respectively in the above staining experiments; therefore, in the selection of time points, this experiment refers to the gocyte nuclear maturation time for statistics; at the same time, it is proved that exogenous ghrelin can accelerate the in vitro maturation of oocytes.

I am sorry, but I still do not fully understand. Did you check the maturation state of both groups after 6h, 8h, 12h, 14h, 16h, 22h? Or if you did it differently in both groups how did you know that you needed to check 2h earlier in the ghrelin group?? For a good comparison you need to compare the different developmental rates at the same time point! Like in table 4 and 5. Here you can see a hugh difference when comparing both groups at 16h! That is impressive!

7.How did you decide how much Ghrelin needs to be used? The difference between 100, 200 and 300ng is not very high.

Response: We thank the reviewer for his/her suggestion. Previous report showed that the addition of 50 ng/mL ghrelin can increase blastocyst rates. However, adding 250 ng/mL ghrelin decreased cleavage and blastocyst rates in sheep's early embryo development1. Thus ,in this study, the use of 100, 200, and 300 ng/mL ghrelin was explored .

Please add this information to the manuscript!

  1. Line 143/154: What is Embryo water? Please define also your stock solutions (in a table maybe)

Response: We thank the reviewer for his/her suggestion. We have added Embryo water and stock solutions instruction.

  1. Statistics: I think a Chi-Share-test would be correct to compare maturation rates between treatment groups and control.

Response: We thank the reviewer for his/her suggestion. We have performed it according to reviewer suggestion.

If so I cannot see it! It is not added to the statistics.

Results

  1. 1B: To me it looks like there cannot be a significant difference between 0; 100 and 300ng treatment. Please check again. Diagram explanation is too small to read.

Line 175: That does not make sense. Do you mean: ….were calculated for the different treatment groups?

Response: We thank the reviewer for his/her suggestion. We have change it to Table3 in revisied version .

Table 3: Maturity – maturation rate!

Table1:Effects of different concentrations of Ghrelin on oocyte formation rate

Concentration of Ghrelin(0ng/mL)

Total number of COCs(Number of repetitions)

Number of oocytes in the MII phase

Maturity rate

0

350(3)

223

63.7%

100

350(3)

253

72.2%*

200

350(3)

300

85.7%**

300

350(3)

256

73.1%*

11.I don´t understand what Fig. 1A should show? I think it is not necessary.

Response: We thank the reviewer for his/her suggestion. We have deleted Figure 1A.

Line 185-188: Is this a Figure legend? What is then Line 195-199?

Fig 1: According to your scale bar your oocytes have now a diameter of about 60µm! I seriously doubt that!

Pictures C: VII and VIII should represent and MI and MII. The nuclear structures are out of focus. Could you present more typical pictures?

Table 4: what are error lines? It is a table! What are the asterisks in the table? Please provide an explanation under the table.

  1. 2: Figures are too small to really read it. Scale bar is probably wrong. Are you sure that the diameter of sheep oocytes is at least 200µm? Human oocytes, for example, are around 100µm.

Response: We thank the reviewer for his/her suggestion. Figure 2 has been divided into three sections to describe and present to the reader.Next, We calculated the number of cells in the experimental and control groups at the time points when the nuclear morphology changed during continuous maturation in vitro. And Figuer2 D is made into a table (Table2) for display, so that readers can see the results more clearly.Finally, we  performed fluorescence quantitative expression detection of cyclin B1 at different stages of oocyte maturation in the experimental group and control group, and changed Figure2E into Figure2 for the reader to consult.

Table 2 Oocyte nuclear maturation changes over time

Control group, Experimental group

  1. I don`t understand why you choose these time points to assess diffenent maturation stages.

Response: We thank the reviewer for his/her suggestion. We have improved the quality of the manuscript with help of many experts.Oocytes can be assessed by changes in nucleus morphology during maturation, and the maturation of oocytes can be intuitively witnessed by staining the nucleus maturation process of oocytes. This experiment determined the time it took for the experimental group and the control group of oocytes to reach the GV, GVBD, MI, and MII stages respectively through continuous live cell staining, and also demonstrated that Ghrelin had an accelerating effect on the in vitro maturation of sheep oocytes.

  1. Fig  3 and 4: too small.

Response: We thank the reviewer for his/her suggestion. Figure 3 has been enlarged and modified as required.inFigure 4 ,In order to make the reader's reading clearer, I have split the GSEA difference comparison chart at different points in time into three pictures, namely: Figure 4: OMM08h vs MGM06h, Figure 5: OMM16h vs MGM14h, Figure 6: OMM22h vs MGM16h.

  1. Fig 4: what is the message?

Response: We thank the reviewer for his/her suggestion. Fig4 was analysis of the Gene set enrichment analysis (GSEA).Emoticons:The green line shows the enrichment profile. NES, normalized enrichment score; FDR, false discovery rate.Figure 4 has been split into three parts according to the difference in the time point of the comparison, and they have been explained separately.

  1. Line 255 ff: Could you provide a table of embryo development? It would be much easier to understand it.

Response: We thank the reviewer for his/her suggestion. We have provided table of embryo development

Table 6: what are error lines? It is a table! What are the asterisks in the table? Please provide an explanation under the table.

Line 307-309: These lines need to be under a different headline

Fig. 7: writing errors of morula and blastocyste

  1. 6: Scale bar is probably incorrect

Response: We thank the reviewer for his/her suggestion. We have revised this figure in revised version.

Discussion

Line 324: what is the oocyte cycle? Please edit.

Line 325: few studies – which? Citation?

Line 335: format needs to be changed, I think.

Line 336: delete marked “the”

Line 341: add space

Line 346-348: You should add here one of your main interesting findings: Ghrelin accelerated IVM significantly!!

Line 353: I don`t understand the meaning of this sentence

Line 363: What is oocytosis? Also, I don`t understand what you want to express.

Line 384: oocyte maturation in vitro instead of oocyte development?

Line 385-388: Long and complicated sentence – please edit.

17.Some parts of the discussion are very detailed but difficult to understand. The authors should define their results more clearly and then focus their discussion on the essential findings. A diagram of the regulatory mechanisms and pathways would be helpful for understanding.

Response: We thank the reviewer for his/her suggestion. We have rewritten discussion with help of many experts.

Appendix

  1. Data Availability Statement: incorrect (hopefully)

Response: We thank the reviewer for his/her suggestion. We have revised it.

Line 400: I cannot see any changes here…

  1. Conflict of interest is actually author contribution, except the last sentence.

Response: We thank the reviewer for his/her suggestion. We have revised it.

Same as before.

  1. The value of the study is severely diminished by the language problems. The manuscript cannot be published in this form. It needs extensive language improvement. This would also help the authors to elaborate their results more precisely and to make the discussion more concise but understandable.

Response: We thank the reviewer for his/her suggestion. We have improved language problems with help of MDPI English editing.

Author Response

We thank the editor and reviewer2 for their enthusiasm and  many insightful comments on our manuscript (1673256). We have comprehensively addressed all of the comments from Reviewer2. We sincerely hope that the revised and improved manuscript is now suitable for publication in Animals Below please find our detailed point-by-point response to the Reviewers.
